# Determination of Whether Apex or Non-Apex Prostate Cancer Is the Best Candidate for the Use of Prostate-Specific Antigen Density to Predict Pathological Grade Group Upgrading and Upstaging after Radical Prostatectomy

**DOI:** 10.3390/jcm12041659

**Published:** 2023-02-19

**Authors:** Cong Huang, Shiming He, Qun He, Yanqing Gong, Gang Song, Liqun Zhou

**Affiliations:** 1Department of Urology, Peking University First Hospital, Beijing 100034, China; 2Institute of Urology, Peking University, National Urological Cancer Center of China, Beijing 100034, China

**Keywords:** prostate cancer, prostate-specific antigen density, apex tumor, Gleason grade group, upgrading, upstaging

## Abstract

**Objective:** Previous studies have demonstrated that prostate-specific antigen density (PSAD) may aid in predicting Gleason grade group (GG) upgrading and pathological upstaging in patients with prostate cancer (PCa). However, the differences and associations between patients with apex prostate cancer (APCa) and non-apex prostate cancer (NAPCa) have not been described. The aim of this study was to explore the different roles of PSAD in predicting GG upgrading and pathological upstaging between APCa and NAPCa. **Patients and Methods:** Five hundred and thirty-five patients who underwent prostate biopsy followed by radical prostatectomy (RP) were enrolled. All patients were diagnosed with PCa and classified as either APCa or NAPCa. Clinical and pathological variables were collected. Univariate, multivariate, and receiver operating characteristic (ROC) analyses were performed. **Results:** Of the entire cohort, 245 patients (45.8%) had GG upgrading. Multivariate analysis revealed that only PSAD (odds ratio [OR]: 4.149, *p* < 0.001) was an independent, significant predictor of upgrading. A total of 262 patients (49.0%) had pathological upstaging. Both PSAD (OR: 4.750, *p* < 0.001) and percentage of positive cores (OR: 5.108, *p* = 0.002) were independently significant predictors of upstaging. Of the 374 patients with NAPCa, 168 (44.9%) displayed GG upgrading. Multivariate analysis also showed PSAD (OR: 8.176, *p* < 0.001) was an independent predictor of upgrading. Upstaging occurred in 159 (42.5%) patients with NAPCa, and PSAD (OR: 4.973, *p* < 0.001) and percentage of positive cores (OR: 3.994, *p* = 0.034) were independently predictive of pathological upstaging. Conversely, of the 161 patients with APCa, 77 (47.8%) were identified with GG upgrading, and 103 (64.0%) patients with pathological upstaging. Multivariate analysis demonstrated that there were no significant predictors, including PSAD, for predicting GG upgrading (*p* = 0.462) and pathological upstaging (*p* = 0.100). **Conclusions:** PSAD may aid in the prediction of GG upgrading and pathological upstaging in patients with PCa. However, this may only be practical in patients with NAPCa but not with APCa. Additional biopsy cores taken from the prostatic apex region may help improve the accuracy of PSAD in predicting GG upgrading and pathological upstaging after RP.

## 1. Introduction

Prostate cancer (PCa) is one of the mostly frequently diagnosed solid malignant tumors in men worldwide [1,2,3]. The treatment options for PCa are generally based upon risk stratification derived from biopsy Gleason score (GS), prostate-specific antigen (PSA), and clinical stage [4]. Thus, biopsy GS and clinical stage are principal factors in the initial assessment of patients with PCa and can inform different therapeutic strategies. Unfortunately, preoperative GS and clinical stage are often inconsistent with final pathological results after radical prostatectomy (RP). Indeed, approximately 30% to 50% of patients experience either GS upgrading or pathological upstaging after analysis of RP specimens [5]. Recently, Epstein et al. proposed an alternative, simplified PCa grading system which is based on the 2005 International Society of Urologic Pathology (ISUP) modified Gleason grading system [6]. This new Gleason grade group (GG) system, which uses the biochemical recurrence of PCa after treatment as a surrogate endpoint to define aggressive disease, appears to improve risk stratification and, consequently, clinical decision making. This grading group system was accepted by the World Health Organization (WHO) for the 2016 edition and has been validated in previous studies [7,8].

Currently, transrectal ultrasound (TRUS)-guided prostate biopsy for clinically suspected PCa detection is the standard of care. However, TRUS-guided biopsy schemes predominantly target the posterior and lateral peripheral regions of the prostate, and therefore it is difficult to sample tumors located in the prostatic anterior apex. Additionally, previous studies found that tumors primarily occurred in the anterior half of the gland at the apex to mid prostate. Both may lead to a higher false-negative rate of transrectal biopsy and increase the risk of GG upgrading and pathological upstaging [9].

The prostate-specific antigen density (PSAD) has been demonstrated to be associated with adverse pathological characteristics and poor prognosis [10,11]. Nonetheless, conflicting results were reported when assessing its ability to predict pathological upgrading and upstaging [12,13]. The controversial results may be due to various confounding factors such as biopsy scheme, tumor volume, or tumor location. To our knowledge, no study has yet compared the accuracy of PSAD in predicting upgrading and upstaging between patients with or without anteriorly apical prostate cancer (APCa).

Thus, the aim of this study was to evaluate the different performance of PSAD as a predictor of prognostic GG upgrading and pathological upstaging between APCa and non-apical prostate cancer (NAPCa).

## 2. Patients and Methods

### 2.1. Patient Selection

The institutional review board approved this retrospective study, and the requirement for informed consent was waived. Between January 2001 and April 2018, the medical records of patients who had received TRUS-guided biopsy resulting in a diagnosis of organ localized PCa (≤T2c) and underwent open, laparoscopic, or robot-assisted RP in our institution within 3 months of diagnosis were retrospectively evaluated. All patients underwent TRUS-guided systematic 12- or 13-core prostate biopsies, with the addition of at least two targeted biopsies at any area suspected of malignancy by ultrasonography. Those who received neoadjuvant androgen deprivation therapy or drugs to alter PSA values were excluded from the study. Patients with incomplete data were also excluded. Ultimately, a total of 535 patients were enrolled in the study.

All RP surgical specimens were fixed in formalin buffer (4%) after the outer surface. Specimens were sliced with standardized multiple transverse cuts, using a modified handling technique described previously by the ISUP Consensus Conference [14]. Notably, the prostatic apex (PA) of RP specimens underwent parasagittal separation and was split into two distal apical 5 mm sections. The patients were classified as either APCa or NAPCa according to the histological examination. APCa was defined as any malignant findings in the PA section, without regard to other locations.

### 2.2. Data Collection

Clinical and pathological data were collected from all the patients. The clinical data included age, body mass index (BMI), serum prostate-specific antigen (PSA), digital rectal examination (DRE), prostate volume (PV) evaluation via TRUS, and clinical T stage (assessed by the 2017 American Joint Committee on Cancer staging system). PSAD was calculated by dividing serum PSA by PV. The pathological data included biopsy and RP specimen GG, number of biopsy cores, number of positive cores, percentage of tumor involvement of each biopsy core, pathological T stage, extracapsular extension, seminal vesicle invasion, positive surgical margin, and lymph node invasion.

Analyses of all needle biopsies and RP specimens were centralized and performed by two dedicated genitourinary pathologists. The overall biopsy GS was based on the core with the highest GS. The overall GS of RP specimens with multifocal lesions was similarly based on the nodule with the highest GS. Gleason grading of prostatic carcinoma followed the 2005 ISUP consensus conference and was adapted to the new Gleason GG system [6]. Upgrading was regarded as an increase from one prognosis GG to another. Upstaging was defined as the pathological diagnosis of non-organ localized disease which was not clinically suspected before RP.

### 2.3. Statistical Analysis

Quantitative variables were described as means ± standard deviation (SD) or medians with their respective interquartile range (IQR), and differences between groups were analyzed using Student’s *t* test or the Mann–Whitney *U* test, as appropriate. Qualitative variables were presented as frequencies and percentages, and differences were compared using chi-square tests. Univariate and multivariate logistic regression analyses were performed to evaluate the independent, significant variables in the prediction of GG upgrading and pathological upstaging. Receiver operating characteristic curves (ROC) were generated to assess the predictive accuracy. Statistical analyses were performed using SPSS version 20.0 (IBM Corporation, Armonk, NY, USA and MedCalc version 18.11 (MedCalc Software, Mariakerke, Belgium). All tests were two-sided and a *p* < 0.05 was considered statistically significant.

## 3. Results

### 3.1. Baseline Characteristics

The baseline clinical and pathological characteristics of the study cohort are shown in Table 1. The median age was 67 years (IQR: 62–71); the median PSA value was 10.93 ng/mL (IQR: 7.49–17.16); the median PV was 39.1 mL (IQR: 30.0–57.0); and the median PSAD was 0.26 ng/mL^2^ (IQR: 0.16–0.44). In addition, the median number of biopsy cores was 13 (IQR: 12–14).

Overall, 161 patients (30.1%) were identified as having APCa. Patients presenting with APCa showed higher preoperative PSA (*p* < 0.001), lower BMI (*p* = 0.005), and higher PSAD (*p* = 0.022). Notably, patients with APCa were more likely to harbor unfavorable clinicopathological features such as a higher percentage of positive cores (*p* < 0.001), higher max core involvement (*p* < 0.001), higher post-RP GG (*p* < 0.001), higher pathological T stage (*p* < 0.001), positive surgical margin (*p* < 0.001), and extracapsular extension (*p* < 0.001). However, there were no significant differences in age (*p* = 0.590), DRE (*p* = 0.228), PV (*p* = 0.175), number of biopsy cores (*p* = 0.360), seminal vesical invasion (*p* = 0.136), and lymph node metastasis (*p* = 0.346).

### 3.2. The Entire Cohort

Of the entire cohort, 245 patients (45.8%) presented with GG upgrading after RP. Patients with GG upgrading had higher serum PSA value (*p* < 0.001) and higher PSAD (*p* < 0.001) compared with those who did not display GG upgrading. No significant differences were found in age (*p* = 0.205), BMI (*p* = 0.418), PV (*p* = 0.119), number of biopsy cores (*p* = 0.430), percentage of positive cores (*p* = 0.599), and max core involvement (*p* = 0.393) (Appendix A). After univariate and multivariate analysis, only PSAD (odds ratio [OR]: 4.149, *p* < 0.001) was found to be an independent, significant predictor of GG upgrading (Table 2).

There were 262 patients (49.0%) who had pathological upstaging after RP. However, no significant differences in age (*p* = 0.359), BMI (*p* = 0.110), or number of biopsy cores (*p* = 0.809) were observed (Appendix A). The univariate analysis showed that higher PSA (OR: 1.035, *p* < 0.001), smaller PV (OR: 0.980, *p* < 0.001), higher PSAD (OR: 7.244, *p* < 0.001), higher number of positive cores (OR: 1.232, *p* < 0.001), higher percentage of positive cores (OR: 15.821, *p* < 0.001), and higher max core involvement (OR: 1.018, *p* < 0.001) were predictive of pathological upstaging. The multivariable analysis revealed that both PSAD (OR: 4.750, *p* < 0.001) and percentage of positive cores (OR: 5.108, *p* = 0.002) were independent, significant predictors of upstaging (Table 3).

### 3.3. The Patients with NAPCa

Of the 374 patients with NAPCa, 168 (44.9%) had GG upgrading after RP. Serum PSA (*p* = 0.001) and PSAD (*p* < 0.001) were significantly higher in upgraded patients than in non-upgraded patients. There were no significant differences in age (*p* = 0.506), BMI (*p* = 0.423), PV (*p* = 0.068), number of biopsy cores (*p* = 0.416), number of positive cores (*p* = 0.414), percentage of positive cores (*p* = 0.610), and max core involvement (*p* = 0.324) (Appendix A). The univariate analysis showed that serum PSA (OR: 1.021, *p* = 0.046), PV (OR: 0.991, *p* = 0.035), and PSAD (OR: 5.429, *p* < 0.001) were significant predictors of GG upgrading. The multivariate analysis showed that PSAD (OR: 8.176, *p* < 0.001) was an independent predictor of GG upgrading (Table 2).

Pathological upstaging occurred in 159 (42.5%) patients with NAPCa. Upstaged patients had higher PSA (*p* < 0.001), smaller PV (*p* < 0.001), higher PSAD (*p* < 0.001), a higher percentage of positive cores (*p* < 0.001), and higher max core involvement (*p* = 0.007). No statistically significant differences were found in age (*p* = 0.694), BMI (*p* = 0.293), or number of biopsy cores (*p* = 0.574) (Appendix A). Univariate analysis revealed that higher PSA (OR: 1.034, *p* = 0.007), lower PV (OR: 0.973, *p* < 0.001), higher PSAD (OR: 7.142, *p* < 0.001), higher number of positive cores (OR: 1.238, *p* < 0.001), higher percentage of positive cores (OR: 15.651, *p* < 0.001), and higher max core involvement (OR: 1.017, *p* < 0.001) were predictive of upstaging. In the multivariate analysis, PSAD (OR: 4.973, *p* = 0.001) and percentage of positive cores (OR: 3.994, *p* = 0.034) were independently predictive of pathological upstaging (Table 3).

### 3.4. The Patients with APCa

Of the 161 patients with APCa, 77 (47.8%) were identified as having GG upgrading after RP. Serum PSA (*p* = 0.024) was significantly higher in patients with GG upgrading than those who did not present GG upgrading. No statistically significant differences were found in age (*p* = 0.215), BMI (*p* = 0.647), PV (*p* = 0.929), PSAD (*p* = 0.182), number of biopsy cores (*p* = 0.633), percentage of positive cores (*p* = 0.764) and max core involvement (*p* = 0.642) (Appendix A). The univariate and multivariate analysis revealed that age (*p* = 0.171), BMI (*p* = 0.606), PSAD (*p* = 0.462), percentage of positive cores (*p* = 0.658), and max core involvement (*p* = 0.910) were not independently associated with GG upgrading (Table 2).

There were 103 (64.0%) patients with pathological upstaging. Upstaged patients had higher PSA (*p* < 0.001), higher PSAD (*p* = 0.002), higher percentage positive cores (*p* = 0.005), and higher max core involvement (*p* = 0.005) compared with those who did not display upstaging. There were no statistically significant differences in age (*p* = 0.384), BMI (*p* = 0.665), PV (*p* = 0.146) or number of biopsy cores (*p* = 0.237) (Appendix A). Univariate analysis revealed that higher PSA (OR: 1.044, *p* = 0.027), higher PSAD (OR: 4.063, *p* = 0.027), higher number of positive cores (OR: 1.171, *p* = 0.005), higher percentage of positive cores (OR: 8.920, *p* = 0.003), and higher max core involvement (OR: 1.016, *p* = 0.007) were predictors of upstaging. However, there were no independent, significant predictors, including PSAD (*p* = 0.100), for predicting pathological upstaging in multivariate analysis (Table 3).

### 3.5. Predictive Characteristics of PSAD

Of the entire cohort, the AUC value of PSAD for predicting GG upgrading was 0.637 (95% CI: 0.595–0.678, *p* < 0.001). The cut-off value of 0.23 ng/mL^2^ showed a sensitivity of 68.98%, specificity of 53.45%, a positive predictive value (PPV) of 55.59%, and a negative predictive value (NPV) of 67.10%. The AUC value of PSAD for predicting upstaging in all patients was 0.737 (95% CI: 0.698–0.774, *p* < 0.001). A cut-off value of 0.23 ng/mL^2^ showed a sensitivity of 77.86%, specificity of 62.27%, a PPV of 66.45%, and a NPV of 74.56% (Table 4).

Of the 374 patients with NAPCa, the AUC value of PSAD for predicting GG upgrading was 0.670 (95% CI: 0.620–0.718, *p* < 0.001). A cut-off value of 0.17 ng/mL^2^ showed a sensitivity of 85.71%, a specificity of 41.75%, a PPV of 54.53%, and a NPV of 78.19%. The AUC value of PSAD for predicting upstaging in patients with NAPCa was 0.775 (95% CI: 0.729–0.816, *p* < 0.001). A cut-off value of 0.23 ng/mL^2^ showed a sensitivity of 79.25%, specificity of 66.98%, a PPV of 63.95%, and a NPV of 81.37% (Table 4).

## 4. Discussion

It is well established that biopsy GG and clinical T stage contribute the most to estimating the prognosis of PCa [15]. However, pathological GG upgrading and upstaging from biopsy to RP specimens is quite common. According to prior studies, the rate of GG upgrading at RP varies from 30% to 50%, meaning that nearly half of all biopsy sampling does not reflect the overall pathological characteristics of prostate specimens [5,16,17]. Furthermore, Gleason GG upgrading and pathological upstaging have been associated with adverse outcomes, including unfavorable pathological features and biochemical recurrence [10]. In the current study, GG upgrading and pathological upstaging after RP were recorded in 45.8% and 49.0% of patients, respectively. Although the definition of upgrading and upstaging may be different between studies, the current results showed a relatively higher rate than those of other reports. This may be because more than one-third of the patients (34.2%, 183/535) in our study were in intermediate or high-risk groups according to D’Amico classification, and the median PSA value was 10.93 ng/mL. Thus, the patients’ characteristics in this cohort were relatively more aggressive than those in other studies. Furthermore, the lack of multi-parametric magnetic resonance imaging (mpMRI) findings, especially in multifocal tumors, may explain the relatively high proportion of patients with GG upgrading at post-RP, as well as the poor performance of biopsy.

Systematic TRUS-guided prostate biopsy has been widely accepted as a mainstay in the diagnosis of PCa, whether it occurs via the transrectal or transperineal approach [15]. Despite the use of appropriate techniques, this method has been shown to underestimate the presence of malignant disease, with false-negative rates ranging from 20% to 40% [18]. The reasons for this occurrence may differ based on tumor location. Particularly in the apex, the occupied volume is very small, and the angle attained by the transrectal approach might be quite limited. It should be noted that the transrectal approach more easily misses tumors located at the apex region. In the current study, all patients underwent TRUS-guided prostate biopsy through the transrectal route. After analysis of the RP specimens, APCa was found in 30.1% (161/535) of patients. In addition, patients with APCa were associated with adverse pathological characteristics. Ishii et al. reported a 36% rate of PCa located predominantly in the apex, and the frequency increased over time [19]. The current results confirmed this previous finding. However, Sazuka et al. demonstrated that in Japanese patients, the apex was the area of cancer most frequently identified (85%), and the section false-negative rate was 45% for transrectal biopsy [20]. These findings suggested that there may be geographic and racial differences in PCa localization.

It is well known that PSAD was initially introduced to improve the accuracy of the PSA test for PCa screening. Several studies have observed that PSAD is significantly better than PSA alone at predicting adverse pathology and biochemical recurrence after RP [12]. The current results also indicate that PSAD may be an effective predictor of adverse pathological features in the entire study cohort (data not shown). Nonetheless, Jones et al. were unable to demonstrate that PSAD outperformed PSA in assessing early biochemical recurrence [21]. Other studies have reported that PSA is more accurate than PSAD in predicting total tumor volume and biochemical recurrence [22]. The discrepancy between those results and the current study may be due to various factors, including differences in tumor location and biopsy schemes between different studies.

Recently, the National Comprehensive Cancer Network guideline has adopted PSAD as an inclusion criterion for active surveillance (AS) in patients with PCa [23]. Ha et al. also demonstrated that removing PSAD from the AS criterion would significantly increase the rate of pathological upgrading and upstaging [24]. However, the association between PSAD and pathological GG upgrading in patients with PCa still remains elusive. In one study, Brassetti et al. recently proved that PSAD is a valuable predictor of upgrading and upstaging in candidates for surgery or AS [25]. Furthermore, Sim et al. also reported that magnetic resonance imaging-based PSAD > 0.26 ng/mL^2^ could aid in the prediction of postoperative upgrading in patients with low-risk PCa [26]. In addition, the specificity and PPV were both relatively high (84.9% and 83.3%, respectively). Nonetheless, Keefe et al. demonstrated that in PCa with a biopsy-proven GS 3 + 4 = 7, clinicopathological features including PSAD were not significantly related to upgrading or upstaging [27]. Ning et al. did not find a significant correlation between PSAD and upgrading using multivariate analysis [28].

Recently, mpMRI of the prostate has increasingly utilized to diagnosis, staging, and risk stratification of PCa [29]. Several systematic reviews have reported that pooled NPVs in the detection of clinically significant PCa for mpMRI ranged from 88% to 93%, with a consequent optimization of the reduction of unnecessary biopsy or overtreatment [30,31]. It is well documented that including mpMRI in an AS cohort may improve the ability to predict GG upgrading. Mamawala et al. showed that mpMRI was an independent predictive factor for GG upgrading in follow-up AS biopsy [32]. However, Chu et al. demonstrated that mpMRI alone was insufficient to detect GG upgrading on AS, especially among patients with PSAD ≥ 0.15 ng/mL^2^ [33]. Meanwhile, Christiansen et al. reported that PSAD was of clinical importance for predicting GG upgrading in patients with PI-RADS 4–5, whereas for men with PI-RADS 4–5, the probability of upgrading was high, regardless of PSAD [34]. Thus, incorporating mpMRI and other clinicopathological parameters including PSAD may overcome the limitations and improve diagnostic accuracy for prediction upgrading.

In the present study, PSAD was an independent, significant predictor of GG upgrading and pathological upstaging when all patients in the cohort were analyzed. The cut-off value proposed for the prediction of GG upgrading was 0.23 ng/mL^2^, but the performance accuracy of PSAD was unsatisfactory, with an AUC value of 63.7%. The sensitivity, specificity, PPV, and NPV were 68.98%, 53.45%, 55.59%, and 67.10%, respectively, which is inferior compared to other studies. Potential confounders include the disadvantages of the biopsy scheme and tumor location in the prostate, which were likely related to limited efficiency in predicting upgrading. Interestingly, after classifying the cohort into APCa and NAPCa groups based on whether the tumor existed in the apex, PSAD only remained significantly associated with Gleason GG upgrading and pathological upstaging in NAPCa patients and was not significant in patients with APCa. In addition, the AUC value of PSAD for predicting GG upgrading in NAPCa patients was 67.0%, with no significant difference before and after classification. However, the sensitivity, specificity, PPV and NPV increased remarkably after grouping, with values of 85.71%, 41.75%, 54.53%, and 78.19%, respectively.

These results suggest that more attention should be paid to the tumor location, especially with regard to the apex region, which might lead to inaccurate biopsy GG evaluation and incorrect analysis. Men with APCa might not benefit from the use of PSAD to predict GG upgrading and pathological upstaging after RP. One possible reason is that all patients in the cohort did not receive an apex-targeted biopsy in the systematic prostate biopsy, and thus small, aggressive PCa with a higher GG at the apex region might be missed. Several studies have demonstrated that adding apex cores improved the detection rate of clinically significant PCa (GS ≥ 7), particularly in early stage disease [35,36]. Therefore, it is especially important in patients with low-risk PCa who seek less invasive therapy, such as watchful waiting and AS, to additionally target the apex region during systematic biopsy. This may help to precisely select patients for AS protocols. Furthermore, comprehensive consideration of PSAD and cancer location may be more reasonable for patient counseling and clinical decision making. Additional sampling of biopsy cores from the apex region may help improve the accuracy of PSAD in predicting GG upgrading and pathological upstaging after RP.

There are several limitations of this study, including its retrospective design and relatively small number of patients. First, there was no systematic, pathological review of all specimens, although the interobserver variability is well known. Second, all patients analyzed in this study underwent TRUS-guided core biopsies without multi-parameter MRI. Multi-parameter MRI focusing on the prostatic apex was superior to systematic biopsy for identifying adverse APCa [37]. In addition, several studies have demonstrated that MRI targeted fusion biopsy could enhance the diagnostic accuracy of PCa detection in final histopathology, with a lower rate of upgrading than TRUS-guided biopsy [3,28]. In this regard, the rate of PCa detection in the current study could have been underestimated, while the rate of upgraded GG could have been overestimated. Third, our study also has a lack of genomic classifiers such as the Oncotype DX Genomic Prostate Score test, which has been reported to be associated with biopsy upgrading [38,39]. Furthermore, the study focus was primarily on the pathological findings. Biochemical recurrence and PCa-specific mortality were not evaluated; these may be more crucial issues than adverse pathological features for better defining tumor progression.

## 5. Conclusions

PSAD may aid in the prediction of GG upgrading and pathological upstaging in patients with PCa. However, this advantage may only be practical in patients with NAPCa identified after RP. Additional biopsy cores taken from the prostatic apex region may help improve the accuracy of PSAD in predicting pathological GG upgrading and upstaging after RP.

## Figures and Tables

**Table 1 jcm-12-01659-t001:** Baseline characteristics of the study cohort.

	Overall (n = 535, 100%)	NAPCa (n = 374, 69.91%)	APCa (n = 161, 30.09%)	*p* Value
Age, years				0.590
Median (IQR)	67 (62–71)	67 (62–71)	67 (61–72)	
Mean ± SD	66.13 ± 6.64	66.15 ± 6.35	66.07 ± 7.29	
BMI, kg/m^2^				0.005 *
Median (IQR)	24.34 (22.68–26.22)	24.22 (22.31–26.09)	24.78 (23.31–26.78)	
Mean ± SD	24.50 ± 2.87	24.27 ± 2.88	25.04 ± 2.78	
Serum PSA, ng/mL				<0.001 *
Median (IQR)	10.93 (7.49–17.16)	10.20 (7.07–16.28)	12.17 (8.55–20.98)	
Mean ± SD	14.47 ± 12.15	13.20 ± 10.69	17.48 ± 14.64	
DRE, n (%)				0.228
Normal	429 (80.2)	305 (81.6)	124 (77.0)	
Abnormal	106 (19.8)	69 (18.4)	37 (23.0)	
Biopsy GG, n (%)				0.057
1	159 (29.7)	124 (33.2)	35 (21.7)	
2	238 (44.5)	161 (43.0)	77 (47.8)	
3	78 (14.6)	51 (13.6)	27 (16.8)	
4	60 (11.2)	38 (10.2)	22 (13.7)	
Post-RP GG, n (%)				<0.001 *
1	58 (10.8)	50 (13.4)	8 (5.0)	
2	239 (44.7)	176 (47.1)	63 (39.1)	
3	163 (30.5)	94 (25.1)	69 (42.9)	
4	41 (7.7)	32 (8.6)	9 (5.6)	
5	34 (6.3)	22 (5.9)	12 (7.5)	
Prostate volume, mL				0.175
Median (IQR)	39.10 (30.00–57.00)	39.00 (29.35–56.10)	42.00 (31.00–57.50)	
Mean ± SD	46.78 ± 25.33	46.51 ± 26.62	47.40 ± 22.10	
PSAD, ng/mL^2^				0.022 *
Median (IQR)	0.26 (0.16–0.44)	0.25 (0.15–0.42)	0.29 (0.18–0.50)	
Mean ± SD	0.36 ± 0.32	0.35 ± 0.32	0.40 ± 0.33	
Number of biopsy cores				0.360
Median (IQR)	13 (12–14)	13 (12–14)	13 (12–13)	
Mean ± SD	13.24 ± 2.41	13.33 ± 2.53	13.04 ± 2.11	
Number of positive cores				<0.001 *
Median (IQR)	4 (2–6)	3 (2–6)	5 (3–7)	
Mean ± SD	4.46 ± 3.06	4.10 ± 2.91	5.27 ± 3.26	
Percent positive biopsy cores, %				<0.001 *
Median (IQR)	30.77 (15.38–50.00)	25.00 (11.76–46.15)	38.46 (20.00–53.85)	
Mean ± SD	34.13 ± 23.65	31.25 ± 22.44	40.81 ± 25.06	
Max core involvement, %				
Median (IQR)	70.0 (30.0–85.0)	60.0 (30.0–85.0)	85.0 (50.0–85.0)	
Mean ± SD	58.68 ± 30.19	55.66 ± 30.48	65.71 ± 28.37	
Clinical T stage, n (%)				<0.001 *
T1	58 (10.8)	28 (7.5)	30 (18.6)	
T2	477 (89.2)	346 (92.5)	131 (81.4)	
Pathological T stage, n (%)				<0.001 *
T2	273 (51.0)	215 (57.5)	58 (36.0)	
T3	262 (49.0)	159 (42.5)	103 (64.0)	
Postoperative pathology, n (%)				
Positive surgical margin	161 (30.1)	61 (16.3)	100 (62.1)	<0.001 *
Extracapsular extension	274 (51.2)	159 (42.5)	115 (71.4)	<0.001 *
Seminal vesicle invasion	90 (17.0)	57 (15.2)	33 (20.5)	0.136
Lymph nodal metastasis	10 (1.9)	7 (1.9)	3 (1.9)	0.346

NAPCa—non-apex prostate cancer; APCa—apex prostate cancer; IQR—interquartile range; SD—standard deviation; BMI—body mass index; PSA—prostate-specific antigen; DRE—digital rectal examination; GG—grading group; RP—radical prostatectomy; PSAD; prostate-specific antigen density. * statistically significant.

**Table 2 jcm-12-01659-t002:** Univariate and multivariate analysis for predicting GG upgrading.

	Univariate		Multivariate	
	OR (95% CI)	*p* Value	OR (95% CI)	*p* Value
(a) All patients (n = 535)				
Age, years	1.018 (0.992–1.045)	0.165	1.022 (0.996–1.050)	0.103
BMI, kg/m^2^	0.971 (0.915–1.031)	0.333	0.992 (0.933–1.055)	0.794
Serum PSA, ng/mL	1.018 (1.003–1.033)	0.018 *	-	-
Prostate volume, mL	0.987 (0.989–1.000)	0.066	-	-
PSAD, ng/mL^2^	3.164 (1.743–5.744)	<0.001 *	4.149 (2.151–8.001)	<0.001 *
Number of biopsy cores	0.978 (0.925–1.034)	0.432	-	
Percent positive biopsy cores, %	0.734 (0.356–1.513)	0.403	0.488 (0.185–1.285)	0.146
Max core involvement, %	0.997 (0.991–1.003)	0.307	0.997 (0.989–1.004)	0.346
(b) Non-apex prostate cancer (n = 374)				
Age, years	1.015 (0.982–1.048)	0.383	1.016 (0.982–1.051)	0.365
BMI, kg/m^2^	0.968 (0.902–1.040)	0.377	0.999 (0.928–1.077)	0.988
Serum PSA, ng/mL	1.021 (1.000–1.043)	0.046 *	-	-
Prostate volume, mL	0.991 (0.983–0.999)	0.035 *	-	-
PSAD, ng/mL^2^	5.429 (2.378–12.397)	<0.001 *	8.176 (3.288–20.331)	<0.001 *
Number of biopsy cores	0.970 (0.904–1.041)	0.401	-	
Percent positive biopsy cores, %	0.639 (0.255–1.597)	0.338	0.371 (0.105–1.305)	0.122
Max core involvement, %	0.996 (0.989–1.003)	0.229	0.995 (0.986–1.003)	0.228
(c) Apex prostate cancer (n = 161)				
Age, years	1.026 (0.982–1.071)	0.250	1.031 (0.987–1.078)	0.171
BMI, kg/m^2^	0.969 (0.866–1.084)	0.578	0.970 (0.866–1.088)	0.606
Serum PSA, ng/mL	1.014 (0.992–1.036)	0.230	-	-
Prostate volume, mL	1.000 (0.986–1.014)	0.970	-	-
PSAD, ng/mL^2^	1.332 (0.516–3.436)	0.554	1.468 (0.528–4.079)	0.462
Number of biopsy cores	0.981 (0.892–1.079)	0.697	-	
Percent positive biopsy cores, %	0.811 (0.235–2.802)	0.741	0.699 (0.143–3.421)	0.658
Max core involvement, %	0.999 (0.988–1.010)	0.843	0.999 (0.986–1.013)	0.910

OR—odds ratio; BMI—body mass index; PSAD—prostate-specific antigen. * statistically significant.

**Table 3 jcm-12-01659-t003:** Univariate and multivariate analysis for predicting pathological upstaging.

	Univariate		Multivariate	
	OR (95% CI)	*p* Value	OR (95% CI)	*p* Value
(a) All patients (n = 535)				
Age, years	1.013 (0.987–1.039)	0.334	1.014 (0.986–1.043)	0.329
BMI, kg/m^2^	1.049 (0.988–1.113)	0.118	1.063 (0.996–1.134)	0.064
Serum PSA, ng/ml	1.035 (1.017–1.053)	<0.001 *	-	-
Prostate volume, ml	0.980 (0.972–0.988)	<0.001 *	-	-
PSAD, ng/mL^2^	7.244 (3.556–14.755)	<0.001 *	4.750 (2.259–9.984)	<0.001 *
Number of biopsy cores	1.232 (1.158–1.311)	<0.001 *	-	
Percent positive biopsy cores, %	15.821 (7.011–35.700)	<0.001 *	5.108 (1.854–14.074)	0.002 *
Max core involvement, %	1.018 (1.012–1.024)	<0.001 *	1.007 (1.000–1.015)	0.051
(b) Non-apex prostate cancer (n = 374)				
Age, years	1.009 (0.977–1.043)	0.568	1.009 (0.972–1.046)	0.645
BMI, kg/m^2^	1.040 (0.968–1.117)	0.287	1.056 (0.977–1.142)	0.171
Serum PSA, ng/mL	1.034 (1.009–1.059)	0.007 *	-	-
Prostate volume, mL	0.973 (0.962–0.984)	<0.001 *	-	-
PSAD, ng/mL^2^	7.142 (3.031–16.830)	<0.001 *	4.973 (1.996–12.391)	0.001 *
Number of biopsy cores	1.238 (1.146–1.336)	<0.001 *	-	
Percent positive biopsy cores, %	15.651 (5.733–42.727)	<0.001 *	3.994 (1.108–14.399)	0.034 *
Max core involvement, %	1.017 (1.009–1.024)	<0.001 *	1.006 (0.997–1.015)	0.199
(c) Apex prostate cancer (n = 161)				
Age, years	1.021 (0.977–1.067)	0.352	1.022 (0.976–1.070)	0.361
BMI, kg/m^2^	1.015 (0.904–1.141)	0.800	1.038 (0.918–1.175)	0.551
Serum PSA, ng/mL	1.044 (1.005–1.085)	0.027 *	-	-
Prostate volume, mL	0.989 (0.975–1.004)	0.142	-	-
PSAD, ng/mL/cm^3^	4.063 (1.169–14.120)	0.027 *	2.906 (0.816–10.342)	0.100
Number of biopsy cores	1.171 (1.048–1.307)	0.005 *	-	
Percent positive biopsy cores, %	8.920 (2.105–37.807)	0.003 *	3.489 (0.576–21.142)	0.174
Max core involvement, %	1.016 (1.004–1.028)	0.007 *	1.009 (0.994–1.023)	0.236

OR—odds ratio; BMI—body mass index; PSAD—prostate-specific antigen. * statistically significant.

**Table 4 jcm-12-01659-t004:** The predictive characteristics of PSAD for predicting upgrading and upstaging.

	AUC (95% CI)	Cutoff, ng/mL^2^	Sensitivity (%)	Specificity (%)	PPV (%)	NPV (%)
(a) All patients (n = 535)						
Upgrading	0.637 (0.595–0.678)	0.23	68.98	53.45	55.59	67.10
Upstaging	0.737 (0.698–0.774)	0.23	77.86	62.27	66.45	74.56
(b) Non-apex prostate cancer (n = 374)						
Upgrading	0.670 (0.620−0.718)	0.17	85.71	41.75	54.53	78.19
Upstaging	0.775 (0.729−0.816)	0.23	79.25	66.98	63.95	81.37

PSAD—prostate-specific antigen; AUC—area under the curve; CI—confidence interval; PPV—positive predictive value; NPV—negative predictive value.

## Data Availability

Not applicable.

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
