# Peer review of "Determination of Whether Apex or Non-Apex Prostate Cancer Is the Best Candidate for the Use of Prostate-Specific Antigen Density to Predict Pathological Grade Group Upgrading and Upstaging after Radical Prostatectomy"

_jcm, 2023, doi:10.3390/jcm12041659_

Round 1

Reviewer 1 Report

Upgrading of Gleason Grading Group (GGG/GG) compared to diagnostic biopsy is a common phenomenon observed in patients following radical prostatectomy. Besides a worse prognosis with increased GG, it may also have a significant impact on treatment outcomes, since e.g., in patients with GG5 radical prostatectomy may be inferior to a combination of teletherapy+brachytherapy+ADT. Therefore developments of methods that increase the precision of pathological grading are of crucial importance in clinical practice. 

In their manuscript, Huang C et al. evaluated the utility of PSAD for the determination of the risk of tumor upstaging and upgrading following biopsy. PSAD added precision in the prediction of final tumor differentiation and local disease extent. However, it was only true for non-apex localized prostate tumors. This is important information for practicing urologists that may help them in the decision-making process regarding optimal local treatment. 

The discussion must include a paragraph on mpMRI utility in the proper determination of tumor grade and stage, which may overcome the limitations associated with PSAD. It not only improves the biopsy procedure but can also stratify prostate cancer patients into low- and high-GGs.

Additional comments

- the Authors mix Gleason score (GS) and Gleason Grading Group (GG/GGG). GS should not be used anymore and they should stick to GGG or GG not GS. They cannot use GS while discussing GGG like in

- paragraph 3.1 "higher biopsy GS (p<0.019), higher RP GS..."

- paragraph 3.4 "...were not independently associated with GS upgrading.

- Authors used GS multiple times in the discussion section - should be changed to GGG/GG

Paragraph 3.3 The sentence should be corrected to: Univariate analysis revealed that higher PSA (...) lower PV (...) and higher PSAD (...) were predictive for upstaging. 

Paragraph 3.4 The sentence should be corrected to: Univariate analysis revealed that lower PV... 

Paragraph 3.5

PSADT is ng/ml not ng/ml2 (should be corrected in paragraph 3.5, discussion [2x])

Table 1 - pathologic GGG should be changed to post-RP GGG

Reviewer 2 Report

In this work the Authors investigate the potential role of age, BMI, PSA, PSAD and prostate volume for the prediction of GS upgrade or pathological upstaging. Interestingly, they also repeat the analyses stratifying the patients for having apex or non-apex PCa, finding differences across the two conditions.

The research question is highly interesting, and the results are worth of note. However, there are methodological issues that question the validity of the present work.

Major comments:

1)      All patients underwent TRUS-guided systematic biopsy, while EAU-EANM-ESTRO-ESUR-ISUP-SIOG guidelines strongly recommend performing mpMRI prior to prostate biopsy. This has been mentioned in the discussion “all patients analyzed in this study underwent TRUS-guided core biopsies without multi-parameter MRI. Multi-parameter MRI focus on prostatic apex was superior to systematic biopsy for iden-tifying adverse APCa”, but still represents a great issue, as this may explain the relatively high proportion of patients with GS upgraded at prostatectomy as well as the poor performance of biopsy, especially for apex PCa.

2)      Why considering patients with GGG = 5 in analyses predicting GS upgrade? These patients are likely to have high PSA levels but their grade cannot upgrade, therefore introducing a huge bias.

3)      Table 3: In univariate analysis prostate volume seems to be a predictor of pathological upstaging in apex PCA, why this variable has been excluded from multivariate analysis?

Minor comments:

1)      Page 5, “The multivariable analysis revealed that only PSAD (OR: 10.826, p < 0.001) was an independent, significant predictor of upstaging (Table 2).” It is table 3, not 2.

Round 2

Reviewer 2 Report

The manuscript can now be accepted for publication.